# Natural Strategies as Potential Weapons against Bacterial Biofilms

**DOI:** 10.3390/life12101618

**Published:** 2022-10-17

**Authors:** Syeda Tasmia Asma, Kálmán Imre, Adriana Morar, Mirela Imre, Ulas Acaroz, Syed Rizwan Ali Shah, Syed Zajif Hussain, Damla Arslan-Acaroz, Fatih Ramazan Istanbullugil, Khodir Madani, Christos Athanassiou, Alexander Atanasoff, Doru Morar, Viorel Herman, Kui Zhu

**Affiliations:** 1Department of Food Hygiene and Technology, Faculty of Veterinary Medicine, Afyon Kocatepe University, Afyonkarahisar 03200, Turkey; 2Department of Animal Production and Veterinary Public Health, Faculty of Veterinary Medicine, University of Life Sciences “King Michael I” from Timișoara, 300645 Timisoara, Romania; 3Department of Parasitology and Dermatology, University of Life Sciences “King Michael I” from Timișoara, 300645 Timisoara, Romania; 4Department of Animal Nutrition and Nutritional Diseases, Faculty of Veterinary Medicine, Afyon Kocatepe University, Afyonkarahisar 03200, Turkey; 5Department of Chemistry and Chemical Engineering, SBA School of Science & Engineering (SBASSE), Lahore University of Management Sciences (LUMS), Lahore 54792, Pakistan; 6Department of Biochemistry, Faculty of Veterinary Medicine, Afyon Kocatepe University, Afyonkarahisar 03200, Turkey; 7Department of Food Hygiene and Technology, Faculty of Veterinary Medicine, Kyrgyz-Turkish Manas University, Bishkek KG-720038, Kyrgyzstan; 8Centre de Recherche en Technologies Agro-Alimentaires, Campus Universitaire Tergua Ouzemmour, Bejaia 06000, Algeria; 9Laboratory of Entomology and Agriculture Zoology, Department of Agriculture, Crop Production and Rural Environment, University of Thessaly, 38446 Volos, Greece; 10Department of Animal Husbandry, Faculty of Veterinary Medicine, Trakia University, Students’ Campus, 6015 Stara Zagora, Bulgaria; 11Department of Internal Medicine, Faculty of Veterinary Medicine, University of Life Sciences “King Michael I” from Timișoara, 300645 Timisoara, Romania; 12Department of Infectious Disease and Preventive Medicine, Faculty of Veterinary Medicine, University of Life Sciences “King Michael I” from Timișoara, 300645 Timisoara, Romania; 13National Center for Veterinary Drug Safety Evaluation, College of Veterinary Medicine, China Agricultural University, Beijing 100193, China

**Keywords:** microbial biofilms, antimicrobial resistance, natural plants, bee products, phytonantechnology

## Abstract

Microbial biofilm is an aggregation of microbial species that are either attached to surfaces or organized into an extracellular matrix. Microbes in the form of biofilms are highly resistant to several antimicrobials compared to planktonic microbial cells. Their resistance developing ability is one of the major root causes of antibiotic resistance in health sectors. Therefore, effective antibiofilm compounds are required to treat biofilm-associated health issues. The awareness of biofilm properties, formation, and resistance mechanisms facilitate researchers to design and develop combating strategies. This review highlights biofilm formation, composition, major stability parameters, resistance mechanisms, pathogenicity, combating strategies, and effective biofilm-controlling compounds. The naturally derived products, particularly plants, have demonstrated significant medicinal properties, producing them a practical approach for controlling biofilm-producing microbes. Despite providing effective antibiofilm activities, the plant-derived antimicrobial compounds may face the limitations of less bioavailability and low concentration of bioactive molecules. The microbes-derived and the phytonanotechnology-based antibiofilm compounds are emerging as an effective approach to inhibit and eliminate the biofilm-producing microbes.

## 1. Introduction

A biofilm is a complex of micro-organisms that sustains a structured and organized pathway for their growth, proliferation, and survival on any surface [1,2]. A single bacterial species may develop the biofilm organizations, or the biofilm can also be of mixed microbial species adhered to a surface [3]. The survival of biofilm-forming bacterial cells depends upon the alignment of extracellular polymeric substance-encapsulated micro-colonies in the matrix [4]. The biofilm-forming microbial cells maintain their micro-environment by controlling temperature, nutrients, and pH, which can affect biofilm formation. Biofilms are known to cause several infections or diseases in humans [5,6]. Several antibiotics have been used against biofilm-associated infections, but increased antimicrobial resistance (AMR) has directly been linked to biofilm microbes [7]. This inefficacy of numerous antibiotics against several biofilm-linked infections has gradually enhanced the emergence of AMR.

The development of microbial resistance to antibiotics reduces or inhibits the efficacy of antibiotics. Consequently, it has been demonstrated that the improper use of antibiotics leads to the emergence of antibiotic resistance [8]. Many antibiotics are losing efficacy due to several micro-organisms’ expeditious developments in multidrug resistance. Several factors responsible for causing AMR and intrinsic biofilm formation have been recognized as major critical factors [9]. Moreover, AMR caused by biofilm development may cause harmful recurrent chronic microbial infections. Therefore, the discovery of a significant treatment approach is much needed to combat AMR. Several researchers are designing and evaluating new combat strategies based on one health concept [10].

This study explores the new combat approaches that avoid existing resistance mechanisms. Using different natural products, such as plant-derived components, bee products, marine-derived components, and plant-based nanomaterials, is gaining great attention from researchers to design novel antibiofilm compounds to avoid AMR. In the current review, the emerging global concerns regarding the development of biofilm resistance and their control by describing some potential therapeutic compounds have been recapitulated.

## 2. Biofilm Formation

Biofilm can be defined as the complex aggregation of micro-organisms, firmly adhered to a surface and micro-colonized into extracellular polymeric substances (EPS) matrix. EPS is comprised of exopolysaccharides, nucleic acids, lipids, and proteins [11]. The emergence of biofilm formation is a multistep process in which EPS performs particularly required functional and structural roles. Microbial communities may attach to both abiotic and biotic surfaces facilitated by EPS. In order to maintain a biofilm lifestyle, the EPS matrix provides essentially required chemical microenvironments and mechanical stability [12].

In addition, the EPS also improves biofilm tolerance toward different antimicrobial agents and immune cells. The biofilm maturation needs several developmental phases with particular features, resulting in biofilm development on the surfaces. Additionally, understanding every developmental phase is essential for designing and applying proper antimicrobial agents against microbial biofilms. Each developmental phase is briefly elaborated on in Figure 1.

### 2.1. Surface Adhesion

Biofilm development is initiated with the attachment of microbial cells to any surface. Several sensing pathways (e.g., BasSR, BaeSR, and CpxAR) facilitate the microbes’ identification of the surface [13]. After initial adhesion, the microbial cells start to multiply and divide expeditiously under favorable conditions [14]. The microbial cells produce adhesins (enzymes) that facilitate their attachment to the host surfaces. This phase explicitly targets the adherence mechanism between the microbes and the surfaces. Thus, in this phase, significant combating intervention can be achieved by entirely disrupting the attachment mechanisms of micro-organisms with the surfaces via mainly targeting cell surface-linked adhesins. Consequently, the initial biofilm development can be inhibited by disrupting the initial adhesion process [15].

Surface adhesion is essentially the required parameter for biofilm development. The adhesion initiation and the biofilm dispersal start with the adherence capacity of a specific bacterial species concerning the host surface. The surface attachment, along with biofilm development, is the survival strategy of the microbes that typically adhere to themselves in a particular way to meet their environmental and nutritional requirements. The surface adhesion consists of two phases: the primary/reversible phase and the secondary/irreversible attachment phase [16]. Both of these phases are entirely controlled by the gene’s expression. The attachment processes require several points to be considered, including microbial species, environmental conditions, gene products, and surface composition [17]. In the reversible phase, microbes hydrophobically interact with the abiotic surfaces for adhesion, whereas adherence with biotic surfaces takes place by developing molecular interactions [18].

### 2.2. Biofilm Maturation

The formation of mature biofilm is followed by early biofilm development when the microbes begin to multiply and divide, creating micro-colonies incorporated with the EPS matrix [19]. The EPS matrix plays a multifunctional role, allowing the establishment of several physical and chemical microhabitats facilitating the microbes to build social and polymicrobial interactions. The established biofilms can be disrupted or removed by employing several targeting approaches, including the EPS matrix disruption, targeting microenvironments (such as hypoxia or low pH), physical removal, targeting polymicrobial interactions, and eliminating dormant cells, etc., providing a great scope for designing biofilm combating antimicrobial therapeutics [20].

### 2.3. Biofilm Dispersal

During the dispersal phase, the micro-organisms of sessile biofilm begin to disperse and change into motile form. On the other hand, the microbes that do not produce extracellular polysaccharides directly scatter into the environment by applying a mechanical force. The dispersed microbial communities produce saccharolytic enzymes, which release surface microbes towards a new place for colonization. The *Escherichia coli* produces N-acetyl-heparosan lyase and *Pseudomonas aeruginosa*, *P. fluorescens* produces alginate lyase, and *Streptococcus equisimilis* produces hyaluronidase. The development of flagella originates with the upregulation of protein expression by microbial cells. It allows the microbes to move to a new place for colonization and supports them in spreading biofilm-associated diseases. The remodeling of the EPS matrix and the dispersal pathways activation may induce dispersion of biofilm that can assist in combating biofilms [18].

## 3. Biofilm Composition

Biofilm is a complex of variability and heterogeneity consisting of 10–25% microbial cells and a 75–90% self-developed EPS matrix [21]. Moreover, the water channels or interstitial voids of microbial biofilms are mandatory for micro-colonies’ separation from each other [22]. The EPS creates a covering scaffold that grips the biofilm cells together and facilitates communication between cell-to-cell, providing cohesive and adhesive forces for biofilm development. EPS facilitates nutrient availability, maintaining the deoxyribonucleic acid (DNA) availability for horizontal gene transfer and provides a defensive barrier against antibiotics, desiccation, oxidizing biocides, host defense immune system, and ultraviolet radiation [23]. The EPS constituents include polysaccharides, extracellular proteins, extracellular DNA, lipids, surfactants, and water.

### 3.1. Polysaccharides

Polysaccharides developed by microbes can be categorized into two categories: hetero-polysaccharides and homo-polysaccharides. Most polysaccharides are heterogeneous in nature, while only a few are homogenous in nature, such as glucans, sucrose-based fructans, and cellulose [24]. Several interactions, such as electrostatic interactions, van der Waals forces, ionic interactions, and hydrogen bonding, help in polysaccharide interactions with themselves or ions and proteins to maintain the architecture of biofilms [25]. The primary function of polysaccharides is to provide a protective role by mediating microbial adhesion among the micro-colonies and maintaining the structural stability of biofilms [26]. Three types of exopolysaccharides, such as alginate, Pel, and Psl, contribute to biofilm development and maintain the structural stability of *P. aeruginosa* biofilms [27]. Some polysaccharides identified in different microbial biofilms are represented in Table 1.

### 3.2. Extracellular Proteins

The biofilm complex can have a substantial number of extracellular proteins [34]. Their interaction with exopolysaccharides and nucleic acids facilitates surface colonization and stabilization in the biofilm matrix [35]. Some proteins mediate the degradation and dispersion of biofilm matrix, such as glycosyl hydrolase, dispersin B induces polysaccharide degradation [36], proteases dissolve the proteins of matrix [37], and some DNases cause extracellular nucleic acid breakage [38]. Toyofuku et al. described that 30% of EPS-matrix proteins in *P. aeruginosa* were observed in outer-membrane vesicles as membrane proteins. Some of them were secreted and lysed by cell-derived proteins [39].

Several extracellular enzymes have also been found in microbial biofilms; some of them are involved in biopolymer degradation. The extracellular enzyme substrates contain water-insoluble components (such as lipids, cellulose, and chitin), water-soluble compounds (such as proteins, polysaccharides, and nucleic acids), and the biofilm entraps organic particles [40]. Additionally, some extracellular enzymes can be used for EPS structural degradation to mediate the microbial detachment in biofilms.

### 3.3. Extracellular DNA (eDNA)

eDNA is one of the prime components of the EPS matrix, which is essential for the accumulation of microbes within the biofilm. The amount of eDNA production may vary even among closely linked microbial species. eDNA is the primary structural constituent in the matrix of a *Staphylococcus aureus* biofilm, while in *S. epidermidis* biofilms, it is produced as a minor constituent [41]. It has been revealed that eDNA is a vital component of the biofilm matrix and its mode of life [42,43]. eDNA has also been found as a major component in the biofilm matrix of *P. aeruginosa* and facilitates intercellular interactions [44].

Moreover, the eDNA was observed to inhibit the *P. aeruginosa* biofilm formation, while in *Bacillus cereus* the eDNA acts similar to adhesins [45,46]. Okshevsky and Meyer reported that eDNA is involved in cell adhesion, structural stability maintenance, and horizontal gene transfer and also protects the immune system and antimicrobials [47]. eDNA has been observed to facilitate cell adhesion and biofilm development in *Listeria monocytogenes* [48]. Wilton et al. found that eDNA causes acidification of the biofilm matrix and consequently enhances resistance of *P. aeruginosa* biofilms against different antibiotics [49].

### 3.4. Surfactants and Lipids

The extracellular polysaccharides, proteins, and eDNA are quite hydrated (hydrophilic) molecules, while the other components of EPS exhibit hydrophobic properties. Some bacterial species, such as *Rhodococcus* spp. generate hydrophobic EPS, and can attach to Teflon and may colonize the waxen surfaces with the help of hydrophobic EPS [50]. In the EPS matrix, a few lipids exhibiting surface active properties, such as surfactin, viscosin, and emulsan, increase the availability of hydrophobic molecules by causing their dispersion [24]. An important surfactant class, biosurfactants, begins the micro-colony formation, aids in biofilm structural integrity, and mediates biofilm dispersal [51].

### 3.5. Water

Water is recognized as the largest constituent (accounts for up to 97%) of the EPS matrix of most of the biofilms, and it retains the biofilm hydrated and protects against desiccation [24]. The water in the biofilm matrix may exist in the form of solvents or can also be bound inside the bacterial cells’ capsules [52]. The binding and movement of water inside the biofilm matrix are essential to diffusion mechanisms that occur inside the biofilm and result in fine biofilm structure development [53]. The amount of available water is responsible for nutrient flow and availability within the microbial biofilms [52].

## 4. Biofilm Structural Stability Parameters

The antimicrobial resistance and the other functional characteristics of microbial biofilms are linked with the structure of the biofilm, matrix shape, and the 3D organization of microbes [54]. The local environmental heterogeneous conditions inside the biofilm matrix impact the microbial gene expression and the metabolic actions of biofilm-developing microbial cells [55,56]. The closely packed microbial cells and the water channels are the two major constituents of biofilm formation [57]. The structural familiarity of microbial biofilms is of greater concern in identifying their behavioral and survival strategies. The biofilm architecture and formation variability have been analyzed by applying specific parameters, such as substratum exposure, bio-volume, thickness, and roughness, and observing significant inter- and intra-species variability [56].

The cell-cell communication, environmental influences, and the secondary messengers, such as c-di-GMP and cAMP, structure the biofilms by providing microbes with better environmental adaptability [58]. Several other factors influence the biofilm architecture, such as nutrient availability, microbial motility, hydrodynamic conditions, exopolysaccharides and protein abundance, and anionic and cationic concentrations within the biofilm. In *P. aeruginosa*, an EPS known as alginate facilitates biofilm formation and its architectural stability [59].

The EPS in *Vibrio cholera* and *E. coli* facilitates biofilm development in a three-dimensional configuration [60,61]. An EPS and a secreted protein called TasA are vital for developing a fruiting body, such as a *Bacillus subtilis* biofilm, and maintaining the biofilm matrix’s integrity [62]. The architecture of biofilm can be altered by the exopolysaccharides’ substituents, such as acetyl groups, known to be responsible for enhanced cohesive and adhesive biofilm properties [24].

### 4.1. Proteins

Manifoili et al. determined the impact of three different mitogen-activated protein kinases (MAPKs) (such as SakA, MpkA, and MpkC) and protein phosphatases (PhpA) on *Aspergillus fumigatus* biofilm formation. MAPKs reduce the *A. fumigatus* adhesion in biofilm formation. The ΔpphA strain was found to be more susceptible to cell wall destroying antimicrobials, had less chitin, and enhanced β-(1,3)-glucan, resulting in reduced adherence and biofilm formation [63].

The bacterial cell wall-linked fibronectin-binding proteins (FnBPs) (FnBPB and FnBPA) mediate the biofilm development of methicillin-resistant *S. aureus* (strain LAC) by promoting bacterial adhesion and accumulation [64]. The outer membrane protein W (OmpW) contributes to *Cronobacter sakazakii* survival and biofilm formation under NaCl-stressed conditions [65]. A surface protein, BapA1, plays a significant role in bacterial adhesin and biofilm formation. BapA1 carries the nine putative pilin iso-peptide linker domains, which are significant for bacterial accumulation of pilus in several Gram-positive bacteria, such as *Streptococcus parasanguinis* [66].

### 4.2. Genes and Signaling Cascades

Intracellular cyclic dinucleotide and extracellular quorum sensing (QS) signaling cascades are crucial in biofilm development. It has been reported that these two signaling pathways may coincide or link up and synergistically mediate biofilm formation [67]. QS is the process of intercellular communication that enables the bacteria to adapt to harsh environmental conditions. They mediate biofilm formation by activating small signaling molecules, such as autoinducer-2 (AI-2), auto-inducing peptide (AIP), and N-acyl-homoserine lactones (AHL), in Gram-positive and -negative bacteria, respectively [68]. AI-2 mediates the QS and biofilm development with *bhp*- and *ica*-dependent modes. AI-2 regulates the QS in *S. epidermidis* through increased transcription levels of *bhp* (biofilm-linked protein containing *icaR*) and *ica* operon [69].

QS signaling cascades observed in *P. aeruginosa* include integrated QS (IQS), PQS, *rhl*, and *las*. These QS systems activate each other by regulating QS-associated genes [70]. The small non-coding RNAs (sRNAs) regulate the bacterial transition from planktonic-sessile bacterial biofilms. The two-component regulatory systems (TCSs), such as RsmZ and RsmY targeting RsmA, are involved in *P. aeruginosa* biofilm formation [71]. Several sRNAs have been studied that regulate the activity or expression of different transcriptional regulators to mediate bacterial adhesion and increase biofilm formation (Table 2).

## 5. Resistance Mechanism in Biofilms

Several antimicrobial resistance mechanisms have been identified, and biofilm development is one of the major factors of resistance emergence. Mechanisms allowing microbial biofilms to resist or tolerate the antimicrobials’ actions are discussed here.

### 5.1. The Structural Complexity of Biofilms

As EPS is crucial for a biofilm’s architectural stability, it also acts as a physical barrier, protecting or shielding the embedded microbes against antimicrobials, ultraviolet light, etc. [85,86]. The polysaccharides (negatively charged) can significantly bind to the aminoglycoside antibiotics (positively charged) and block their penetration [87]. The EPS barrier can reduce the diffusion of small compounds, such as H_2_O_2_ (hydrogen peroxide), to microbial cells inside the biofilm. It has been observed that *P. aeruginosa* in the planktonic state is more susceptible to H_2_O_2_, while in the form of a biofilm, it can survive even at a very high concentration of H_2_O_2_ [88].

### 5.2. The Heterogeneity of Biofilms

The heterogeneity inside the developed biofilms averts the entire eradication of all involved microbial cells by antimicrobials. There are oxygen and nutrient gradients from the top-bottom of microbial biofilms. From the top-bottom of the biofilm matrix, the oxygen and nutrient reduction leads to a reduced growth rate and metabolic activity [89]. The protein expression of bacterial cells in biofilm is diverse and quite different from planktonic cells, which may also contribute to microbial resistance development. For example, in its biofilm form, a rice endophytic bacterium, *Pantoea agglomerans* YS19, expresses SPM43.1 protein (acid-resistant) at high levels to resist harsh environmental conditions [90,91].

### 5.3. Quorum Sensing

Quorum sensing (QS) in microbes is a cell-cell communication mediated by activating specific signaling molecules, facilitating environmental adaptation to microbes [2]. QS is a crucial mechanism for regulating and developing biofilms by reducing or inhibiting the effectiveness of antimicrobials against biofilm bacteria [92]. Gram-negative and positive bacterial species communicate using these signaling molecules, also known as autoinducers (AIs) [2]. Some QS signaling molecules used by Gram-positive and negative bacteria include (a) N-acyl homoserine lactone (AHL), (b) autoinducer-2 (AI-2) by *V. harveyi*, (c) autoinducing peptide 1 (AIP-1) by *S. aureus*, (d) N-(3-oxoacyl)-l-homoserine lactone (3-oxo-AHL), (e) diffusible signaling factor (DSF), (f) N-(3-hydroxyacyl) homoserine lactone (3-hydroxy-AHL), (g) 2-heptyl-3-hydroxy-4(1H)-quinolone by *P. aeruginosa*, and (h) hydroxy-palmitic acid methyl ester (PAME). They are presented in Figure 2.

Environmental factors, such as nutrient deficiency, pH, antimicrobials, and salt concentrations, regulate QS-mediated activity in bacterial biofilms [93]. The feed-forward mechanism enhances the QS communication in the biofilm matrix [94]. The QS signals facilitate biofilm formation when their concentration reaches the threshold [95]. Subsequently, the QS signaling molecules are translated into cells for gene expression modulation. These genes are crucial for environmental adaptation, leading to biofilm formation [96]. The QS signaling system regulates bacterial and EPS secretion systems [97] and multidrug efflux pumps [98]. The complete information about QS signaling molecules may offer significant information for developing novel methods or chemicals for combating microbial biofilms. This research era has grabbed the great attention of researchers for designing and developing significant molecules with the ability to neutralize or compete with the QS signaling molecules or their receptors [99].

### 5.4. Enhanced Efflux Pumps

The efflux pumps (proteinaceous) entrenched in the cytoplasmic membranes act as active transporters. The multidrug and toxic compound extrusion family (MATE), the ATP-binding cassette family (ABC), the small multidrug resistance family (SMR), the resistance-nodulation-division family (RND), and the major facilitator superfamily (MF) are commonly reported efflux pump classes in different bacteria [100,101].

Several molecular studies have revealed that increased efflux pumps are a popular and criticizing resistance mechanism in microbial biofilms [102]. This mechanism has been extensively studied in a commonly found biofilm-producing *P. aeruginosa* pathogen [103]. The PA1874-1877 (cluster of genes) involved in developing resistance in biofilms was discovered by Zhang and Mah. Overexpression of PA1874-1877 in biofilm cells facilitates resistance in a biofilm-specific manner [104]. Numerous efflux pump genes inducing biofilm-specific resistance through their overexpression have been identified. For example, in RND-3 efflux pumps, the overexpression of BCAL1672-1676 induces biofilm resistance against ciprofloxacin and tobramycin, while in RND-8 and RND-9, the overexpression of BCAM0925-0927 and BCAM1945-1947 provides resistance to *Burkholderia cepacia* against tobramycin [105].

## 6. Pathogenicity of Biofilm Microbes

Numerous factors are known to contribute to pathogenicity in biofilm-producing microbes. Microbial biofilms release different extracellular substances, altering the gene regulation of several microbial virulence factors. Moreover, the biofilm-producing microbes strengthen the maturation rate of biofilms to escape from host defenses, enhance the activity of β-lactamase, and for plasmid-mediated gene transfer resulting in intense virulence and antimicrobial resistance with enhanced mutation rate and efflux pump. The properties of the extracellular matrix contribute to the biofilm’s pathogenicity, offering a defensive barrier with less antimicrobial and immune cell penetration [106].

The MIC of antibiotics is significant against planktonic microbes but not effective against biofilm microbes [107]. Microbial biofilms can induce several persistent biofilm-associated infections, such as urinary tract infections, middle-ear infections, dental caries, endocarditis, cystic fibrosis, osteomyelitis, and implant-induced infections. Numerous pathogenic microbes are involved in causing persistent biofilm infections; some of them are listed in Table 3.

The pathogenic activity of microbes in the form of biofilms is significantly higher, and they can escape from host defense cells and antibiotics. In numerous infections, biofilm bacteria are concerned with the pathogenesis and clinical symptoms [114]. Opportunistic pathogenic bacteria, such as *P. aeruginosa* and *S. aureus*, can cause chronic biofilm infections, and hospitalized individuals (approximately 8–10%) are more vulnerable to carrying infections.

### 6.1. Health Problems and Infections Caused by Biofilm Bacteria

Biofilm-associated infections pose a threat to human health. Over the last few decades, innovative methods have been discovered to control microbial infections. Biofilm formation in the era of the food industry poses a serious threat to human health. Biofilms may contain only one type of bacteria, different bacterial species, or fungal species that may be pathogenic and may only target immunocompromised patients (cancer patients, organ recipients, HIV patients, etc.). Systematic diseases (*E. coli*, *L. monocytogenes*), food intoxication (*P. aeruginosa*, *S. aureus*, *B. cereus*), and gastroenteritis (*Salmonella enterica*, *E. coli*) can be caused by biofilm-producing pathogens [114].

### 6.2. Biofilms in the Food Industry

Foodborne infections may arise from microbial biofilm development on food processing equipment or food matrices. Biofilms formed on food processing equipment can secrete toxins and may result in food poisoning. Biofilm formation in any food industry may put human health at potential risk. The severity of the risk is directly dependent on the microbial species of the biofilm matrix.

Food processing plants provide suitable conditions for biofilm development on food surfaces due to the complexity of manufacturing or processing plants, mass product yield, long manufacturing durations, and large biofilm formation areas [115]. These biofilm formations may contribute to the emergence of biofilm-associated foodborne infections. Approximately 80% of microbial infections in the USA are considered to be specifically associated with biofilm-producing foodborne pathogens [116]. Mixed-species or polymicrobial biofilm formation is a highly diverse phenomenon and depends upon environmental conditions [117], adherence characteristics of surfaces [118], involved microbial cells [119], and components of the food matrix [120]. Adherence surface characteristics, such as electrostatic charge, topography, interface roughness, and hydrophobicity, impact biofilm development and consequently affect the surface’s hygienic status [118,121].

Properties of microbial cells, such as components of cell membranes (e.g., lipopolysaccharides and proteins), hydrophobicity, exopolysaccharides (EPS) production by microbes, and the bacterial appendages (e.g., fimbriae, pili, and flagella), contribute to a crucial role in biofilm formation [118]. Some studies have reported that microbial adherence is more likely to develop on rough surfaces [122], and some experiments indicated no association between microbial adherence and surface roughness [123]. The components of the food matrix in food processing plants may influence microbial adhesion, such as food waste, e.g., carbohydrates, proteins, and fat-enriched meat and milk exudates, mediate microbial growth and proliferation, and facilitate the dual-species biofilm development by *S. aureus* and *E. coli* [124,125]. Biofilm-producing foodborne pathogens have emerged as a serious threat to human health. Some foodborne pathogens with biofilm-forming ability and their harmful effects are listed in Table 4. Biofilms have been found to be associated with different outbreaks or epidemics (Table 5).

## 7. Biofilm Control

The global rise in antibiotic resistance has led to the failure of antibiotics. The ABR has become a major threat to human health. Therefore, alternative therapies have been reported to eliminate or inhibit biofilm formation and their associated infections. Different points can be targeted for biofilm inhibition and eradication at different stages of biofilm formation (Figure 3). These combating strategies include inhibition of planktonic cells, inhibition of bacterial adhesion, surface alteration, biofilm removal, degradation of EPS, QS inhibition, dispersion of biofilms, and matrix degradation.

## 8. Biofilm Controlling Compounds

Many natural compounds can act as biofilm-controlling compounds by interfering with QS, possessing antiadhesive properties, and inhibiting biofilm formation (Figure 4).

### 8.1. Natural Plants and Bee Products

Several naturally occurring compounds can be used as antibiofilm molecules to eradicate or inhibit biofilm development. The garlic extract can significantly block QS and may promote rapid virulence attenuation (e.g., elastase, protease A, exo- and cytotoxin production or motility, and adhesion capacity reduction) of *P. aeruginosa* by polymorphonuclear leukocytes (PMNs) within the immune response of a mouse infection model [146]. Persson et al. synthesized some QSIs derived from garlic extracts and AHLs [147]. *Chamaemelum nobile* is a naturally occurring, well-known plant for its antimicrobial, anti-inflammatory, antiseptic, spasmolytic, anticatarrhal, sedative, and carminative properties. It can inhibit *P. aeruginosa* biofilm by disrupting QS [148]. Proanthocyanidins extracted from cranberries have significantly inhibited the adhesion of *E. coli* to uroepithelial cells [149]. Cranberry juice also significantly inhibited the site-specific adherence of *Helicobacter pylori* to the gastric mucous of humans [150], and it also prevented *Streptococci* spp. biofilm formation [151,152]. Eighty medicinal plants, more prominently *Fritillaria verticillata*, *Rhus verniciflua*, *Cocculus trilobus*, and *Liriope platyphylla*, have been analyzed for their antibiofilm activity. Comparatively, *Cocculus trilobus* (ethyl acetate fraction) has shown the highest antibiofilm activity against Gram-positive bacteria by providing effective antiadhesive activity [153]. A Chinese herb, *Herba patriniae,* with medicinal properties, has averted the gene expression of six genes linked with biofilm development and EPS production in *P. aeruginosa* [154]. The Ginkgollic acid isolated from a plant, *Ginkgo biloba*, exhibited antitumor, antimicrobial, and neuroprotective and is active against *S. aureus* strains and *E. coli* biofilm formation [155,156].

Elekhnawy et al. determined the antiquorum sensing and biofilm inhibitory potentials of *Dioon spinulosum* plant extract against the clinical isolates of *P. aeruginosa*. The in vitro analysis of antibiofilm activity showed a 77.1–34.3% reduction in biofilm formation at 250–500 μg/mL concentrations. Both in vitro and in vivo investigations revealed a significant reduction in *P. aeruginosa* biofilm formation. However, preclinical studies leading to clinical studies are recommended to allow its practical application in treating *P. aeruginosa* infections [157]. Obaid et al. studied the antibiofilm activity of six plant extracts, such as *Apium graveolens*, *Plantago ovata*, *Vitis vinifera*, *Viscus album*, *Senna acutifolia*, and *Melissa officinalis*, against *Aggregatibacter actinomycetemcomitans*. The *A. actinomycetemcomitans* was collected from patients with dental caries. The obtained results indicated that the *A. actinomycetemcomitans* was more sensitive to *S. acutifolia* and *M. officinalis*, with zone inhibitions of 33 and 35 mm, respectively [158]. Negam et al. evaluated the antifungal and antibiofilm potential of *Encephalartos laurentianus* (methanol extract) against *C. albicans*. The in vitro antibiofilm analysis revealed a 62.5–25% reduction in *C. albicans* cell percentage. The in vivo evaluations of *E. laurentianus* performed on *C. albicans* infected rats resulted in an increased survival rate with a protective effect against renal damage caused by *C. albicans* [159].

Olawuwo et al. investigated the in vitro antibiofilm activity of *Acalypha wilkesiana*, *Alchornea laxiflora*, *Ficus exasperata*, *Jatropha gossypiifolia*, *Morinda lucida*, and *Ocimum gratissimum* plant extracts against poultry pathogens (*Aspergillus flavus*, *A. fumigatus*, *C. albicans*, *Campylobacter* spp., *Salmonella* spp., *E. coli*, *S. aureus,* and *Enterococcus faecalis*). All plant extracts showed effective biofilm inhibition of approximately >50% against the tested micro-organisms [160]. Fathi et al. determined the antibiofilm potential of *Malva sylvestris* methanolic extract against some human pathogens, such as *E. coli*, *S. aureus*, *P. aeruginosa*, *E. faecalis*, and *K. pneumoniae*. The highest biofilm inhibition was examined against *S. aureus* (89.19%), *K. pneumoniae* (95.46%), and *E. faecalis* (98.79%) with 40 μg/mL MIC [161].

Priyanto et al. studied the antibiofilm potential of leaf extract of *Paederia foetida* against *E. coli*, *Mycobacterium smegmatis* with 30–50% inhibition, respectively [162]. Panjaitan et al. evaluated the in vitro antibiofilm potential of ethanol extract of *Cinnamomum buramanii* against periodontal pathogens, such as *Aggregatibacter actinomycetemcomitans* and *Porphyromonas gingivalis*. The outcomes revealed that all *C. buramanii* concentrations showed effective antibiofilm activity against both periodontal pathogens [163].

Plescia et al. determined the antibiofilm potential of *Artemisia arborescens* plant extracts against *S. aureus* (ATCC 25923), *E. coli* (ATCC 25922), *P. aeruginosa* (ATCC 15442), *E. faecalis* (29212), and *C. albicans* (10231). The hot methanol extract showed the highest antibiofilm activity against *S. aureus*, *E. coli*, and *C. albicans* with 58–67% inhibition [164]. Rhimi et al. investigated the in vitro antibiofilm activity of EOs of *Cymbopogon* spp. (*Cymbopogon proximus* and *Cymbopogon citratus*) against *Malassezia furfur* and *Candida* spp. The EOs of *C. proximus* and *C. citratus* showed significant biofilm inhibition ranging from 27.65 ± 11.7 to 96.39 ± 2.8 against all the tested organisms. Based on the reported results, the EOs of both *Cymbopogon* spp. can be used for the prevention of *Malassezia* and *Candida* infections [165].

Nazzaro et al. determined the antibiofilm activity of EOs of aerial parts and bulbs of two different cultivars of *Allium sativum* (Bianco del Veneto, Staravec) against nosocomial and food pathogens *S. aureus*, *E. coli*, *L. monocytogenes*, and *Acinetobacter baumannii*. The EOs from the bulbs and aerial parts of Bianco del Veneto showed significant inhibitory activity against all tested bacteria, more prominently against *L. monocytogenes* 64.29–60.55%, respectively. The EOs from the aerial parts of Staravec exhibited effective inhibition more effectively against *Acinetobacter baumannii* (45.61%), while EOs from the bulbs of Staravec showed no inhibition. The outcomes revealed their potential application as potential antibiofilm agents in the food industry and health sector as well [166].

Gamal El-Din et al. investigated the in vitro antibiofilm potential of EOs of three species of *Jatropha* flowering plant. The EOs were obtained from *J. intigrimma*, *J. gossypiifolia*, and *J. roseae*. The 7.81, 15.63, 31.25, 62.5, 125, 250, 500, and 1000 μg/mL concentrations were used to evaluate the antibiofilm activity. *J. intigrimma* EO exhibited 100% biofilm inhibitory activity at 31.25 μg/mL. *J. roseae* EO showed 100% inhibition at 250 μg/mL, while *J. gossypiifolia* EO revealed less effective biofilm inhibition even at 1000 μg/mL. However, it can be suggested that *J. intigrimma* and *J. roseae* EOs can be used as promising antibiofilms and furthering in vivo investigations is also highly recommended [167]. Djebilli et al. determined the composition profile, antioxidant, and antibiofilm efficacy of EOs from Algerian aromatic plants, including *Thymus algeriensis*, *Eucalyptus globulus*, and *Origanum glandulosum*. The EOs from all three plants showed significant antibiofilm activity with low minimum inhibitory concentrations (MICs) ranging between 0.078–1.25 μg/mL against *C. albicans*, *E. coli*, *E. faecalis*, *L. monocytogenes*, *S. aureus*, and *S.* Typhimurium. The order of biofilm inhibition against the tested bacteria was revealed as *T. algeriensis* > *O. glandulosum* > *E. globulus* EOs. The *T. algeriensis* EO showed the highest inhibition against *L. monocytogenes* (80.95%), and *E. coli* (77.83%) at MICs [168]. Some other biofilm inhibiting plant extracts and essential oils are listed in Table 6.

Bee products are effectively being studied for their wide range of antibacterial, antioxidant, antiviral, antifungal, and anticancerous activities. Honey and its bioactive components are well recognized for their potential antibacterial effects against a wide range of bacteria and even against several antibiotic-resistant bacteria [185,186,187].

Bouchelaghem et al. collected propolis from six Hungarian regions and evaluated the in vitro antibiofilm activity by using ethanolic extract (EEP) alone and in combination with vancomycin against MSSA and MRSA. The EEP significantly prevented planktonic growth. The EEP in combination with vancomycin synergistically showed effective inhibition and degradation against biofilm formation and maturation, respectively. The EEP at a concentration of 200 μg/mL against MSSA and MRSA showed 47–87% biofilm degradation, respectively [188]. Alandejani et al. have determined the antibiofilm activity of four different kinds of honey: Manuka honey (from New Zealand), Buckwheat and Canadian clover honey (from Canada), and Sidr honey (from Yemen). All these honey samples have shown considerable bactericidal activity against *P. aeruginosa*, methicillin-resistant *S. aureus* (*MRSA*), and methicillin-sensitive *S. aureus* (*MSSA*) biofilms. Manuka and Sidr honey have notably more effective antibiofilm activities against *P. aeruginosa*, *MSSA*, and *MRSA*, ranging from 91%, 63–82%, and 63–73%, respectively [189]. Manuka honey has also shown effective results against some Gram-negative bacteria, including extended-spectrum β-lactamase (ESBL) and carbapenemase-producing *K. pneumoniae* [190,191,192], ESBL-producing *E. coli* [191,193], multidrug-resistant (MDR) *P. aeruginosa* [191], antibiotic-resistant *Ureaplasma urealyticum*, and *Ureaplasma parvum* [194].

Fadl et al. reported the antibiofilm potential of bee venom against biofilm-forming MDR bacteria, such as *S. aureus*, vancomycin-resistant *S. aureus* (VRSA), *P. aeruginosa*, *Enterobacter cloacae*, and *S. haemolyticus*. Bee venom showed a considerable reduction in biofilm formation, ranging between 63.8–92% [195]. Bouchelaghem et al. determined the anti-biofilm impact of Hungarian propolis. The ethanolic extract of propolis (EEP) was used to evaluate the antibiofilm effect against MSSA and MRSA by applying a crystal violet assay. The EEP alone and in combination with vancomycin were tested against MSSA and MRSA. The EEP significantly inhibited biofilm development and degraded MSSA and MRSA mature biofilms. The EEP, combined with vancomycin, synergistically enhanced the antibiofilm activity against MRSA [188].

### 8.2. Nanotechnology and Phyto-Nanotechnology

Nanotechnology-based nanomaterials (NMs) are very small in size (˂100 nm) with a large surface area and may provide several biological, chemical, and biomedical applications [196]. NMs can easily enter into the outer membrane (EPS) to release the antimicrobials to targeted sites without damage. Several NM types have been designed and evaluated to inhibit or eradicate microbial biofilms. Nanoparticles are mainly categorized into two categories: organic nanoparticles (NPs) (including polymers, cyclodextrins (CDs), liposomes, dendrimers, and solid lipid NPs) and inorganic NPs (including metal oxides, quantum dots, metallic NPs, and fullerene) [197]. NMs provide potential microbicidal activity alone or in combination with encapsulated drugs [198]. The NMs are reported to provide a promising therapeutic potential for developing significant antibiofilm action [199]. The development of plant-derived nanoparticles (NPs) has emerged as an innovative approach in the era of nanotechnology by synthesizing environmentally friendly substances with little to no toxicity [200]. Several researchers have significantly reported the green synthesis of NPs and their antimicrobial potential against different bacterial biofilms.

Swidan et al. investigated the biofilm inhibition of Ag NPs against enterococcal clinical isolates of the urinary tract (biofilm producing). Three types of Ag NPs were prepared to investigate the antibiofilm activity, including cinnamon Ag NPs synthesized by *Cinnamon cassia*, ginger Ag NPs synthesized by *Zingiber officinale*, and the chemically synthesized Ag NPs. The outcomes demonstrated that the chemical and ginger Ag NPs decreased the biofilm formation to 65.32% and 39.14%, and the adhesion to the catheter surface to 69.84% and 42.73%, respectively, and the cinnamon Ag NPs were not as significant. The ginger Ag NPs showed the most effective antibacterial and antiadhesion effects against the enterococcal clinical isolates that can also produce biofilms [201]. Muthulakshmi et al. synthesized the Ag NPs using *Terminalia catappa* plant leaves and evaluated the in vitro and in vivo antibiofilm potential against the foodborne pathogen *L. monocytogenes*. The in vitro analysis showed 33–45.5% biofilm inhibition at 50–100 μg/mL, respectively. The in vivo evaluation of Ag NPs using *Caenorhabditis Elegans* revealed 90% antiadherent activity against *L. monocytogenes* [202]. Salem et al. synthesized and characterized selenium NPs with orange peel. The biosynthesized Se NPs were used to evaluate the antibiofilm potential against *S. aureus*, *K. pneumonia*, and *P. aeruginosa*. The Se NPs at 0.5 μg/mL concentration showed 95, 88, and 75.5% biofilm inhibition against *K. pneumonia*, *S. aureus*, and *P. aeruginosa*, respectively [203]. Some plant-based NPs, along with their biofilm inhibitory actions, are summarized in Table 7.

#### 8.2.1. Liposomes

Liposomes can significantly enhance the interaction with bacterial membranes and mediate penetration into mature biofilms due to their high biocompatibility. The fusogenic liposome can be significantly employed to encapsulate drugs or antimicrobial agents with optimized release to increase the antibiofilm efficacy (Figure 5). Vancomycin encapsulated with fusogenic liposomes (able to fuse with bacterial membranes) enhanced the bactericidal efficacy against *S. aureus* biofilms [213]. Meropenem-loaded nanoliposome with a dosage of ≥1.5 µg/mL entirely eradicated the *P. aeruginosa* biofilm [214]. The drug encapsulated in liposomes showed more effective results than the pristine drug. Liposomes as nanocarriers allow optimized drug release and inhibit biofilm growth.

#### 8.2.2. Solid Lipid NPs

Solid lipid NPs (SLNs) containing surfactants stabilized lipid cores are spherical colloidal NMs (with 10–1000 nm diameters) [215]. Cefuroxime axetil-containing SLNs (CA-SLNs) were designed and analyzed against *S. aureus* biofilms. CA-SLNs showed a two-fold higher anti-biofilm activity against *S. aureus* biofilm. Ninety-seven percent of the free CA was released in 2 h, while it takes 12 h to reach 96% release when it is in the encapsulated form [216].

#### 8.2.3. QS Inhibiting NMs

Numerous QS inhibitor (QSI) NMs have been developed to inhibit or eradicate biofilm formation [217]. These QSI-NMs have numerous benefits over conventional QSIs. NMs can penetrate the biofilm matrix due to their smaller size and high solubility, providing optimized and targeted drug release. A QSI entrapped inside solid-lipid NPs with an ultra-small size (<100 nm diameter) exhibits a 7-fold higher anti-biofilm activity against *P. aeruginosa* than a free QSI [217]. Tellurium (Te) and selenium (Se) NPs have been designed and tested against *P. aeruginosa* biofilms. Se-NPs and Te-NPs disrupted QS signaling and significantly reduced the bacterial biovolume, resulting in biofilm formation inhibition of 70–80% [218]. However, they were observed to be less effective in removing mature biofilms.

#### 8.2.4. Cyclodextrins

Nguyen et al. investigated the anti-biofilm activity of a ubiquitous flavanone, naringenin (NAR), encapsulated β-CD and chitosan against the biofilm of *E. coli* [219]. NAR nanoencapsulation showed an effective antibiofilm potential against *E. coli*. Miconazole encapsulated CDs attached to polypropylene and polyethylene showed 87–96% inhibition against the biofilm formation of *C. albicans* [220]. CDs coated with drugs covalently bind to the surfaces and effectively inhibit *C. albicans* biofilm. Thymol and anidulafungin encapsulated cyclodextrins showed significant antibiofilm activity against the biofilms of *C. albicans* with 75–64% inhibition, respectively [221].

#### 8.2.5. Hydrogels

Many hydrogels have been developed to inhibit or eradicate several microbial biofilms. Some of these, such as metal nanoparticle-based hydrogels [222,223], chitosan-based hydrogels [224,225], bovine serum albumin (BSA) protein-based hydrogels [226], and pentapeptide-based supramolecular hydrogels [227], have exhibited significant biofilm inhibitory and eradication activities against several bacterial biofilms [228]. Some bacteriophages [229,230], drugs [231], and different antimicrobials or active compounds [232] have also been encapsulated into hydrogels, providing efficient anti-biofilm activities with the optimized release of active compounds against different multidrug-resistant bacteria, such as *P. aeruginosa*, *S. aureus*, *Acinetobacter baumannii*, etc. Hydrogels can be used as promising biomaterials to treat and control multidrug-resistant bacterial infections. Some other NMs have also been reported for combating bacterial biofilms (Table 8).

### 8.3. Microbes and Marine-Derived Anti-Biofilm Compounds

Many micro-organisms produce several types of bioactive molecules with anti-microbial properties to benefit from other micro-organisms. Several investigations have determined the various secondary metabolites with potential anti-biofilm properties extracted from different bacterial and fungal species. Streptomyces species are known as the most promising sources of biofilm-controlling compounds. Methanolic compounds extracted from *Streptomyces* sp. (strain MUC 125) showed potential anti-biofilm activity against *MRSA* due to its 2,3-dihydroxybenzoic acid-mediated iron chelating ability [260]. Ethyl-acetate secondary metabolites extracted from *Streptomyces* sp. (isolated from Iraqi marine sediment) exhibited significant potential for designing and developing new anti-biotics for treating urinary tract infections. The extract showed biofilm inhibition against *P. mirabilis* (uropathogenic bacteria) even at sub-MIC by interrupting the QS signals [261].

Marine-derived bacteria are well categorized as a prime microbial group found in the marine environment. Peach et al. discovered and characterized the auromomycin chromophore as a potential inhibitor against *V. cholera* biofilms at 60.1 µM (IC_50_). Additionally, the inhibitory effect of auromomycin was significantly enhanced by adding some antibiotics (such as tetracycline, ciprofloxacin, and ceftazidime) at their sub-inhibitory concentrations [262]. A red algae-halogenated furanone, *Dilsea pulchra*, has been reported to have effective anti-biofilm properties [263]. Some synthetic and natural brominated furanones have been reported as significant QS inhibitors against gram-negative and positive bacterial species [264,265,266,267].

Pereira et al. determined the biofilm inhibitory effect of brominated alkylidene lactams (compounds **1**–**6**, Figure 6) against *S. mutans*, *S. epidermidis*, *S. aureus*, and *P. aeruginosa* biofilms [268]. The compound-**1**, γ-hydroxy-γ-lactam, and the compound-**2**, (E)- γ-alkylidene- γ-lactam were found to be most effective against *S. epidermidis* with IC_50_ values of 13.3 and 12.2 µg/mL, respectively. The compound-**3** and **4**, and (Z)-γ-alkylidene-γ-lactam were observed to be most significant against *P. aeruginosa* with IC_50_ values of 0.7 and 0.6 µg/mL, respectively. The compound-**5** was most effective against *S. aureus* with 53.1% biofilm inhibition at 44 µg/mL. The compound-**2**, **4**, and **6** inhibited the biofilm formation of *S. mutans* [268]. Antibiofilm activity of natural and synthetic cembranoid compounds has been analyzed against *P. aeruginosa*, *V. harveyi*, *S. aureus*, and *Chromobacterium violaceum* [269].

*Actinobacteria* are gram-positive and the most versatile bacteria in nature, ranging from aerobic_anaerobic, motile_nonmotile, and sporing_nonsporing bacteria [270]. *Actinobacterial* spp. produces several secondary metabolites that can act as antibacterial, antiviral, antifungal, and anticancer agents. Some of the recently reported antibiofilm actinobacteria are listed in Table 9. Song et al. revealed that some bacterial strains derived from coral *Pocillopora damicornis* showed anti-biofilm activity by QS inhibition. Predominantly, H12-*Vibrio alginolyticus* (a coral symbiotic bacteria) inhibited the biofilm formation of *P. aeruginosa* PAO1 by interfering with *rhl* and the *las* system [271].

Chen et al. isolated three bioactive compounds (benzyl benzoate, 2-methyl-N-(2’-phenylethyl)-butyramide, and 2-methyl-N-(2’-phenylethyl)-butyramide) from a marine bacterium *Oceanobacillus* sp. (XC22919) and analyzed their antibiofilm activity. All three compounds exhibited significant biofilm inhibitory activity against *P. aeruginosa* biofilm in a dose-dependent manner by inhibiting QS activity [272].

## 9. Methodology

### Data Collection Criteria

A bibliographic search was conducted to screen relevant scientific publications before August 2022, on bacterial biofilm formation, resistance development mechanisms, and biofilm control strategies. The online search engines of Google scholar, PubMed, ScienceDirect, and Web of Science Core Collection databases were used. The search strategy consisted of separate or simultaneous use, under different combination forms, of several particular keywords including “biofilm formation”, “microbial biofilm composition”, antimicrobial resistance”, “plant-derived anti-biofilm compounds”, “bee products”, “marine derived compounds”, and “phyto-nanotechnology”. Firstly, all titles and abstracts of the search results were individually examined to evaluate whether the articles met inclusion criteria, meaning reporting results in evaluating the antibiofilm activity of plant products, bee products, and nanomaterials against biofilm-forming bacteria with significant outcomes. All the duplicated papers and those published in a language other than English or irrelevant publications that did not contribute to retrieving meaningful results in the goal of assessing natural strategies as potential weapons against bacterial biofilms were excluded. Furthermore, the selected articles were entirely read to obtain significant material based on their experimental outcomes.

## 10. Conclusions

Over the last three decades, biofilm formation has become a potential threat in the food and health sector. Many biofilm-forming microbial species can develop resistance to harsh environmental conditions. Several antibiotics and different disinfectants are used in hospitals and the food industry. Several biofilm-forming foodborne pathogens have been found to cause outbreaks. Several chronic infections are associated with biofilm-forming microbes. Therefore, several strategies have been designed and analyzed to inhibit microbial growth. However, the emergence of antimicrobial and multidrug resistance forced researchers to study all growth features of microbes and their resistance mechanisms for developing effective combating strategies and significant biofilm-controlling compounds. Many researchers are discovering and developing effective anti-biofilm compounds using natural products. In this review, recent studies were reviewed, focusing on biofilm-controlling compounds, such as natural plants, bee products, nanomaterials, microbes, and marine-derived anti-biofilm compounds to reduce or eradicate microbial biofilms and their associated infections as well. These antibiofilm compounds possess a significant potential to overcome the antimicrobial resistance linked to biofilm formation. On the basis of their potential antibiofilm properties with less to no toxicity and increased bioavailability, they can be suggested to treat biofilm-associated infections.

## Figures and Tables

**Figure 1 life-12-01618-f001:**
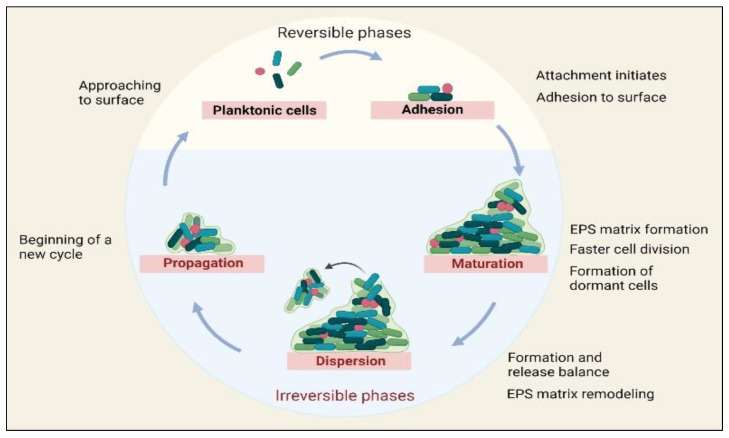
The development phases of biofilms.

**Figure 2 life-12-01618-f002:**
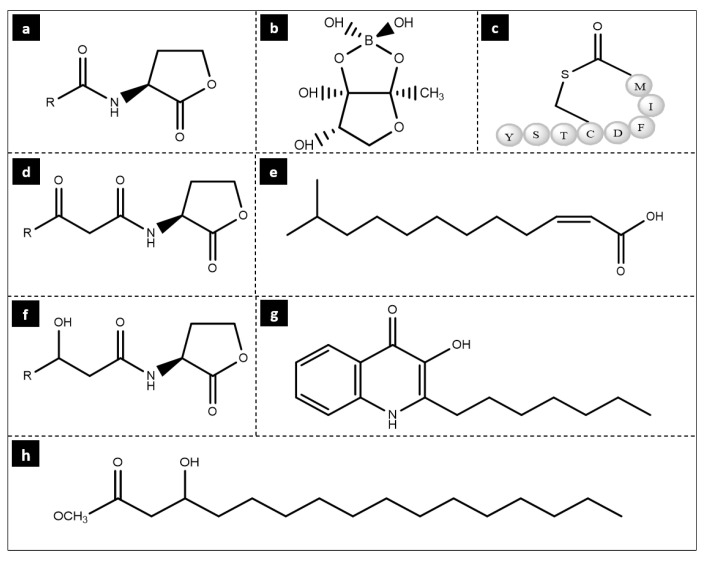
Chemical structures of some signaling molecules.

**Figure 3 life-12-01618-f003:**
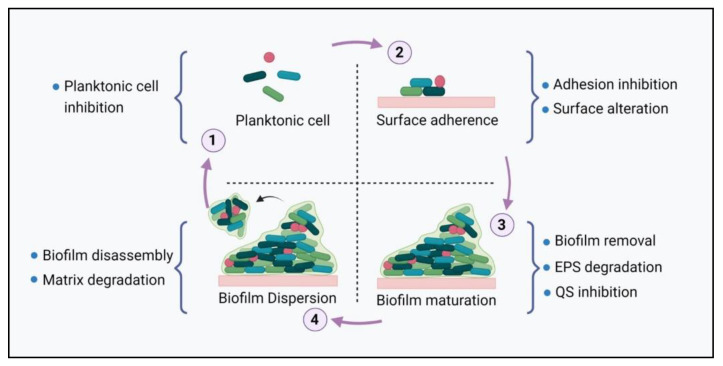
The life cycle of biofilm formation provides different intervention points for biofilm inhibition and eradication.

**Figure 4 life-12-01618-f004:**
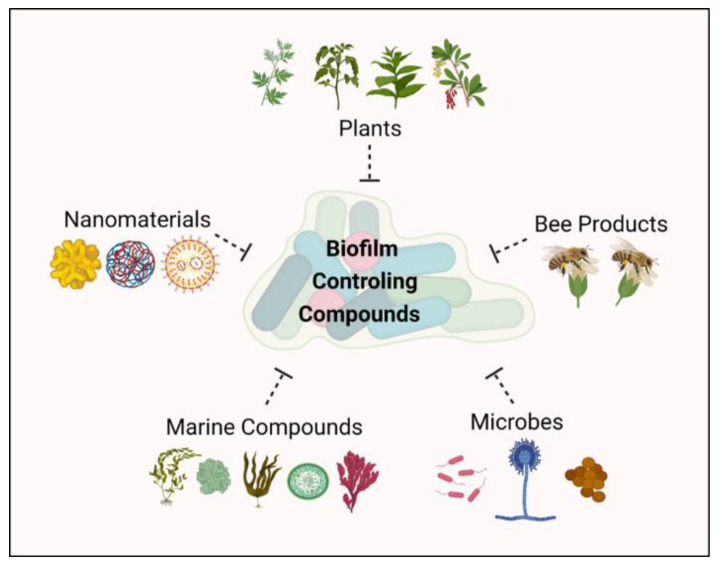
Compounds that can inhibit biofilm formation.

**Figure 5 life-12-01618-f005:**
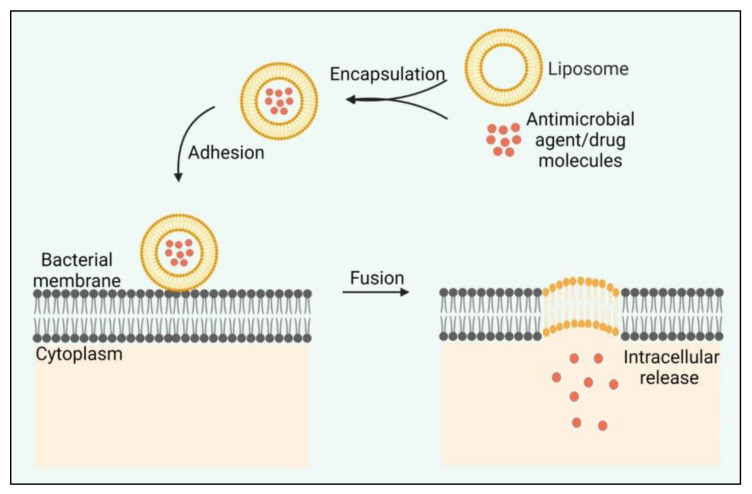
Schematic illustration of the mode of action of the fusogenic liposome.

**Figure 6 life-12-01618-f006:**
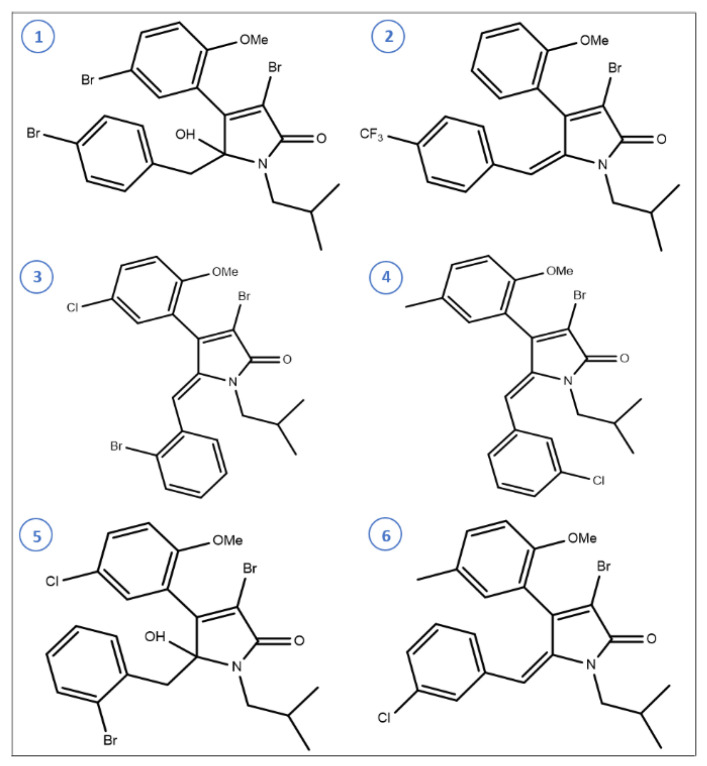
Chemical structures of some brominated alkylidene lactams.

**Table 1 life-12-01618-t001:** List of a few identified polysaccharides in microbial biofilms.

Polysaccharides	Microbes	Role	Ref.
PGA (Poly-β-1,6-N-acetyl-D-glucosamine)	*Actinobacillus actinomycetemcomitans*	Intercellular adhesion Cellular detachment Dispersion	[28]
Colanic acid	*E. coli*	Antidesiccative	[29]
Galactopyranosyl-glycerol-phosphate	*Bacillus licheniformis*	Antibiofilm	[30]
Alginate, Pel, Psl	*P. aeruginosa*	Biofilm structural stability maintenance Cell communication and differentiation	[31,32]
Capsular polysaccharide, cellulose	*Salmonella* Typhimurium	Adhesion Environmental survival	[33]

**Table 2 life-12-01618-t002:** Some identified sRNAs with their targets, bacteria, and role in the regulation of target mRNA.

Targets	sRNAs	Bacteria	Regulatory Effect	Refs.
AphA, HapR	Qrr1-4, Qrr1-5	*V. cholera*, *V. harveyi*	Activation	[72]
Crc	CrcZ	*Pseudomonas* spp.	Repression	[73]
CsgD	GcvB	*E. coli*	Repression	[74]
	McaS	*E. coli*	Repression	[75]
	OmrA, OmrB	*E. coli*	Repression	[76]
	SdsR	*S.* Typhimurium	Activation	[77]
CsgD, YdaM	RprA	*E. coli*	Repression	[78]
CsgD, RpoS	ArcZ	*S. Typhimurium*, *E. coli*	Activation	[77,79,80]
CsrA	CsrB, CsrC	*Yersinia pseudotuberculosis, E. coli*	Repression	[81,82]
PgaA	McaS	*E. coli*	Activation	[74]
PqsR	PhrS	*P. aeruginosa*	Activation	[83]
RpoS	DsrA	*E. coli*	Activation	[80]
	OxyS	*E. coli*	Repression	[80]
	RprA	*E. coli*	Activation	[84]

**Table 3 life-12-01618-t003:** Biofilm-producing pathogenic microbes involved in causing infections.

Pathogenic Microbes	Targeted Area	Consequences	Refs.
Group A streptococci	Skin	Necrotizing fasciitis, tissue necrosis	[108]
*Actinobacillus*, *Actinomycetemcomitans*, *Eikenella corrodens*, *Streptococcus mutans*, *Prevotella intermedia*, *Porphyromonas gingivalis*, Oral spirochetes.	Oral cavity	Periodontal infections, acute inflammation, teeth loosening due to periodontal tissue breakdown, halitosis	[109,110]
*Streptococcus* spp. *Staphylococci* (coagulase-negative), *S. aureus*, *Enterococcus* spp.	Musculoskeletal system	Bacterial accumulation on implants and dead bones cause biofilm infections.	[111]
*Haemophilus influenza*, *Streptococcus pneumoniae*, *Moraxella catarrhalis*.	Middle ear	Otitis media	[112]
*S. aureus*, *P. aeruginosa*.	Lungs (in patients with cystic fibrosis)	Mucoviscidosis, lung infections	[113]

**Table 4 life-12-01618-t004:** Biofilm-associated foodborne pathogens along with their consequences.

Organism	Contaminated Food Items	Consequences	Refs.
*Anoxybacillus flavithermus*	Milk powder	Reduced acceptability of powdered milk	[126]
*B. cereus*	Meat, dairy products, vegetables, and rice	Vomiting, diarrhea	[127,128]
*E. coli*	Meat, vegetables, fruits, and milk	Hemolytic uremic syndrome, diarrhea	[114]
*Campylobacter jejuni*	Unpasteurized milk, animals, poultry	Vomiting, bloody diarrhea, nausea, fever, and stomach cramps	[129]
*S. enterica*	Porcine, bovine, fish, ovine, and poultry meat	Septicemia, gastroenteritis	[130]
*Geobacillus stearothermophilus*	Dairy dried products	Enzymes or acids production resulting in off-flavors	[130]
*Pseudomonas* spp.	Meat, vegetables, fruits, and dairy products	Blue discoloration occurrence on fresh cheese	[131]
*S. aureus*	Dairy products, poultry, eggs, meat, salads, cakes, and pastries	Diarrhea, vomiting	[132,133]
*L. monocytogenes*	Ready-to-eat products, raw milk, dairy products,	Listeriosis in immune-compromised, elderly, and pregnant patients	[134]
*Pectinatus* spp.	Brewery environment and beer	Produces beer turbid due to the production of sulfur compounds	[135]

**Table 5 life-12-01618-t005:** List of some outbreaks caused by biofilm-associated foodborne pathogens.

Region and Year	Reported Cases	Responsible Organisms	Food Type	Ref.
South Africa (2017–2018)	1060	*L. monocytogenes*	Ready-to-eat meat products	[136]
England (2018)	34	*Clostridium perfringens*	Cheese sauce	[137]
England (2015)	NA	*E. coli* O157:H7	Prepacked salad leaves	[138]
Massachusetts (2014–2018)	1200 per year	*Salmonella*	NA	[139]
England (2016)	69	*Campylobacter*	Raw milk	[140]
Belgium (2013)	52	*S. aureus*	Several foods	[141]
China (2010–2014)	1040	*Vibrio parahaemolyticus*	NA	[142]
Europe (2007–2014)	6657	*B. cereus*	NA	[143]
China (2003–2008)	9041	*V. parahaemolyticus*	Meat and aquatic products	[144]
Australia (2001–2010)	667	*L. monocytogenes*	NA	[145]

NA—Not available.

**Table 6 life-12-01618-t006:** Antibiofilm activity of some plant extracts and essential oils.

Plant Extracts, Compounds or Essential Oils (EOs)	Plant Source	Bacterial Species	Inhibition Concentration	Biofilm Inhibition (%)	Ref.
Leaf extract	*Cochlospermum regium*	MRSA	2000 μg/mL	100	[169]
5-Hydroxymethylfurfural	*Musa acuminata*	*P. aeruginosa*	10 μg/mL	83	[170]
Syringopicroside	*Syringa oblata*	*Streptococcus suis*	1.28 μg/mL	92	[171]
Xanthohumol	*Humulus lupulus*	*S. aureus*	9.8 μg/mL	100	[172]
Sotetsuflavone	*Cycas media* R. Br	*E. faecalis*	-	60.87–21.74	[173]
Lemon grass EO	*Cymbopogon citratus*	*S. aureus*	250 μg/mL	50	[174]
		*P. aeruginosa*	-		
Citral		*P. aeruginosa*	0.40 μg/mL		
		*S. aureus*	>107 μg/mL		
EO	*Cinnamomum verum*	*Acinetobacter baumanii, Citrobacter freundii* *Corynebacterium striatum, E. coli, Klebsiella spp., S. aureus, Salmonella spp., P. aeruginosa*	10 μg/mL	97	[175]
	*Thymus vulgaris*			88	
	*Eugenia caryophyllata*			91	
Ethanol extract	*Azadirachta indica*	MRSA	2000 μg/mL	43.0	[176]
	*Moringa oleifera*	MRSA	2000 μg/mL	51.4	
	*Murraya koenigii*	MRSA	2000 μg/mL	44.9	
	*Psidium guajava*	MRSA	2000 μg/mL	80	
Petroleum ether extract	*Azadirachta indica*	MRSA	2000 μg/mL	83.8	
	*Moringa oleifera*	MRSA	2000 μg/mL	59.9	
	*Murraya koenigii*	MRSA	2000 μg/mL	63.7	
	*Psidium guajava*	MRSA	2000 μg/mL	62.9	
Aqueous extract	*Acacia nilotica*	*K. pneumoniae*	13,300 μg/mL	59	[177]
		*E.coli*	13,300 μg/mL	63	
		*P. aeruginosa*	15,000 μg/mL	39	
		*Proteus mirabilis*	16,700 μg/mL	49	
EO	*Cinnamomum zeylanicum*	*E. coli*	2000 μg/mL	82.76	[178]
		*S. epidermidis*		83.33	
	*Citrus grandis*	*E. coli*	2000 μg/mL	58.62	
		*S. epidermidis*		46.67	
	*Citrus hystrix*	*E. coli*	2000 μg/mL	75.86	
		*S. epidermidis*		83.33	
	*Citrus reticulata*	*E. coli*	2000 μg/mL	82.76	
		*S. epidermidis*		83.33	
	*Psiadia argute*	*E. coli*	2000 μg/mL	90	
		*S. epidermidis*			
	*Psiadia terebinthina*	*E. coli*	2000 μg/mL	93.67	
		*S. epidermidis*		90	
Vanilic acid	*Vaccinium macrocarpon* Aiton (Cranberry)	*E. coli*	23.78 mM	100	[179]
Protocaterchuic			25.95 mM		
Catechin			55.12, 68.9 mM		
Pulp extract	*Euterpe oleracea*	*S. aureus*	250 μg/mL	100	[180]
Leaf extract	*Juglans regia*	*P. aeruginosa*	16,000 μg/mL	60	[181]
Leaf extract	*Tetradenia riparia*	MRSA	-	50	[182]
	*Rosmarinus officinalis*	MRSA	30 μg/mL	50	
Extract	*Tagetes minuta*	*Bacillus* sp. Mcn4	100 μg/mL	50	[183]
Extract	*Tessaria absinthioides*	*Bacillus* spp.	100 μg/mL	66	
		*Staphylococcus* sp. Mcr1	10–50 μg/mL	55–62	
Sesquiterpene lactones	*Acanthospermum hispidum*	*P. aeruginosa*	0.25–2.5 μg/mL	69–77	[184]

**Table 7 life-12-01618-t007:** Biofilm inhibition of some plant-based nanoparticles.

Nanoparticles	Plant Species	Bacteria	Inhibition Concentration	Biofilm Inhibition (%)	Ref.
Ag NPs	*Morinda citrifolia*	*S. aureus*	60 μg/mL	96	[204]
	*Glochidion lanceolarium*	*P. aeruginosa, E. coli, S. aureus*	68.9, 12.9, 23.4 μg/mL	99	[205]
	*Semecarpus anacardium*	*P. aeruginosa, E. coli, S. aureus*	45.5, 23.4, 64.1 μg/mL	>99	
	*Bridelia retusa*	*P. aeruginosa, E. coli, S. aureus*	52.5, 33.8, 32.7 μg/mL		
	*Malus domestica*	*K. pneumoniae*	24.6 μg/mL	34	[206]
		*Enterobacter aerogenes*	35.6 μg/mL	72	
	*Piper betle*	*P. aeruginosa*	8 μg/mL	78	[207]
BER-RHE NPs	*Coptis chinensis* (Berberine), *Rheum palmatum* L. (Rhein)	*S. aureus*	0.1 mmol/mL	96	[208]
CA-BBR NPs	*Coptidis rhizome* (Berberine) *Cinnamomi cortex* (Cinnamic acid)	MRSA	0.1 μmol/mL	64	[209]
Cu NPs	*Cymbopogon citratus*	*E. coli*	2000 μg/mL	49	[210]
		MRSA		33	
	*Crotalaria candicans*	MRSA	1 μg/mL	>75	[211]
ZnO NPs	*Myristica fragrans*	*E. coli*	1000 μg/mL	24	[212]
		MRSA	1500 μg/mL	51	

**Table 8 life-12-01618-t008:** Application of some significant nanomaterials for combating biofilms.

Nanomaterials	Target Organism	Impact on Biofilm	Refs.
Cyclodextrins	*S. aureus, MRSA, C. albicans,* *P. aeruginosa, P. vulgaris,* *E. faecalis, E. coli*	Adhesion inhibition, biofilm eradication	[221,233,234]
Dendrimers	MRSA, MSSA, *E. coli,* *K. pneumoniae, P. aeruginosa*	Biofilm inhibition	[235]
Hydrogels	*P. aeruginosa,* MRSA, *S. aureus, A. baumanii*	Biofilm eradication, wound healing	[225,236,237]
Liposomes	*S. aureus, P. gingivalis*	Growth inhibition, biofilm formation reduction	[238,239]
Polymeric NPs	*E. coli, S. aureus, S. mutans,* *En. Cloacae, P. aeruginosa*	Growth inhibition, matrix disruption, and eradication.	[240,241]
Stimuli-responsive NPs: Ag NPs	*S. aureus, E. coli, P. aeruginosa, S. flexneri,* *K. pneumoniae, S. mutans* *C. albicans, S. aureus, E. coli,* *P. aeruginosa,* *P. aeruginosa, H. pylori,* *S. aureus, S. mutans,* *M. tuberculosis*	Structural alternation, inhibition, oxidative stress.	[236,242,243,244]
Au NPs		Growth inhibition, matrix disruption.	[245,246,247,248,249]
SPIONs		Colonization prevention, cell lysis, oxidative stress	[250,251,252,253,254]
Solid Lipid NPs	*S. aureus*	Growth inhibition	[215,255]
Other inorganic NPs	*S. aureus, P. aeruginosa,* *E. coli, S. epidermidis*	Growth inhibition, matrix disruption	[205,256,257,258,259]

**Table 9 life-12-01618-t009:** Some of the biofilm control compounds of actinobacteria.

Anti-Biofilm Compounds	Source	Target Organism	Biofilm Inhibition	Ref.
Actinobacteria				
Carotenoid pigment	*Streptomyces parvulus*	*C. albicans*	>50%	[273]
Bioactive metabolites	*Frankia casuarinae* DDNSF-02	*Candida* sp. *Pseudomonas* sp.	59–81% 65–80%	[274]
Secondary metabolites	*Streptomyces californicus* Strain ADR1	*S. aureus* and MRSA	90%	[275]
Melanin pigments	*Nocardiopsis dassonvillei* strain JN1, Nocardiopsis sp. JN2	*Staphylococcus* sp.	64.20% (JN1) 65.99% (JN2)	[276]
1-hydroxy-1norresistomycin (HNM)	*Streptomyces variabilis*	*V. cholera* *E. coli* *S. aureus*	92% 96% 93%	[277]
Pyrrolo (1,2-a) pyrazine-1,4-dione hexahydro-3-(2-methylpropyl)	*Actinomycetes Nocardiopsis* sp. GRG 1 (KT235640)	*E. coli* *P. mirabilis*	77% 82%	[278]
Actinomycin-D	*S.* *parvulus*	*S. aureus* *Ruegeria* sp. *P. aeruginosa* *Micrococcus luteus*	53.72% 45.98% 37.12% 22.20%	[279]
Bacterial compounds				
Secondary metabolites	*Streptomyces* (marine sediment)	*P. mirabilis*	63–26%	[261]
N-acyl homoserine lactone-based QS analogs	Aqueous extract of *Rhizobium* sp. NAO1	*P. aeruginosa*	77.9%	[280]
Carolacton	Extract of *Sorangium cellulosum*	*Streptococcus oralis,* *Streptococcus gordonii,* *S. mutans,* *A. actinomycetemocomitans*	Reduced biofilm formation	[281]
CFS	*Clostridium butyricum*	*A. baumannii* (MDR strain)	Dose-dependent biofilm inhibition	[282]
CFS	*Lactobacillus* strains	*P. aeruginosa*	0–64% 100% (with *L. fermentum* L1 and L2)	[283]
CFS	*Lacticasebacillus rhamnosus* GG	*E. coli*	Dose and time-dependent biofilm disruption	[284]
Fungal compounds				
Diterpenoid sphaeropsidin A	*Diploidia corticola*	*P. aeruginosa* (MDR strain) MRSA	62% 53%	[285]
Organic extracts	*Penicillium* sp.	*P. aeruginosa*	Biofilm formation reduction by QS inhibition	[286]
Vulculic acid, curvulol	*Chaetosphaeronema achilleae*	*S. aureus* DSM 1104	91.9–96.8%	[287]
Thiodiketopiperazine derivatives	*Phoma* sp. GG1F1	*S. pyogenes* *S. aureus*	60.7–86% 28–57%	[288]
Crude extract	*Alternaria alternate*	*P. aeruginosa* PAO1	65.2%	[289]
Equisetin	*Fusarium sp.* Z10	*P. aeruginosa* PAO1	58.3%	[290]

CFS—Cell-free supernatant.

## Data Availability

Not applicable.

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
