# Peer review of "Natural Strategies as Potential Weapons against Bacterial Biofilms"

_life, 2022, doi:10.3390/life12101618_

Round 1

Reviewer 1 Report (Previous Reviewer 3)

Authors have adequately responded to the reviewers comments

Author Response

Dear Reviewer,

Thank you again for your contribution in improving the quality of the manuscript!

Reviewer 2 Report (Previous Reviewer 2)

The new version of the manuscript entitled “Naturally derived compounds as potential weapons against bacterial biofilms” has been reviewed and all the responses from the Authors have been carefully checked, trying to do my best in order to provide a "precise" and not a "general" evaluation. The new version has been improved and now there are more evidences of a critical assessment of the data. The new MS better describes a "state of the art" of the topic. Anyway, the methodology is still weak, unclear and difficult to replicate in the light of the description in point 9. Trying to perform a search on PubMed using one of the keywords mentioned at point 9, e.g. 'biofilm formation', 43,139 results are found. If you select articles with the same keywords only in the abstract, the results are 42,712. The same search conducted on Science Direct shows 80,851 results, how much time and human resources did it take to "entirely read" and select all these articles? I suggest to better explain the selection criteria adding a flowchart concerning the search and selection of studies, including all the exclusion criteria which have been applied. The MS has been definitely improved, a further effort to improve the methodology is however appropriate. For these reasons, I suggest to approve the articles with minor revision.

Author Response

Reviewer #2

The new version of the manuscript entitled “Naturally derived compounds as potential weapons against bacterial biofilms” has been reviewed and all the responses from the Authors have been carefully checked, trying to do my best in order to provide a "precise" and not a "general" evaluation. The new version has been improved and now there are more evidences of a critical assessment of the data. The new MS better describes a "state of the art" of the topic. Anyway, the methodology is still weak, unclear and difficult to replicate in the light of the description in point 9. Trying to perform a search on PubMed using one of the keywords mentioned at point 9, e.g. 'biofilm formation', 43,139 results are found. If you select articles with the same keywords only in the abstract, the results are 42,712. The same search conducted on Science Direct shows 80,851 results, how much time and human resources did it take to "entirely read" and select all these articles? I suggest to better explain the selection criteria adding a flowchart concerning the search and selection of studies, including all the exclusion criteria which have been applied. The MS has been definitely improved, a further effort to improve the methodology is however appropriate. For these reasons, I suggest to approve the articles with minor revision.

Dear Reviewer, the authors sincerely thank for the valuable suggestions of the reviewer in necessity of substantial improvement of the “Materials and methods” section. Nonetheless, the authors honestly admit the fact that, during manuscript preparation, they applied a „data collection” criteria (as highlighted in the previous revised version under „9. Methodology” section), rather than a „systematic review”, as it is usually required for review articles. Therefore, presently, the insertion of a flowchart highlighting the exact number of „first database search results”, and their subsequent trimming based on algorithm, which will reflect the number of studies that have been excluded leading to a total of 290 articles (how many are currently in the reference list), seems to be impossible. In addition, dozens of new articles were included in the reference list, as reviewer requirements, during the manuscript revision, However, the authors tried to do their best to improve the „Materials and methods section”, according to the reviewer suggestion, resulting in:

9. Methodology

Data collection criteria

A bibliographic search was achieved to screen relevant scientific publications before August 2022, on bacterial biofilm formation, resistance development mechanisms, and biofilm control strategies. The online search engines of Google Scholar, PubMed, ScienceDirect and Web of Science Core Collection databases were used. The search strategy consisted on separate or simultaneous use, under different combination forms, of several particular keywords including “biofilm formation”, “ microbial biofilm composition”, "antimicrobial resistance”, “plant-derived anti-biofilm compounds”, “bee products”, “marine-derived compounds”, and “phyto-nanotechnology”. Firstly, all titles and abstract of the searches were individually examined to evaluate whether the articles met inclusion criteria meaning reporting results in evaluating the anti-biofilm activity of plant products, bee products, and nanomaterials against biofilm-forming microbes with significant outcomes. All the duplicated and published in other than English language papers, or irrelevant publications that did not conributed in retrieving meaningful results in the goal of assessing natural strategies as potential weapons against bacterial biofilms were excluded. Furthermore, the selected articles were entirely read to obtain significant material based on their experimental outcomes.”

Please read the resulted corrections in the Materials and methods section of the revised version of the manuscript (lines 692-714)!

Thank you again for your efforts!

Yours faithfully!

Reviewer 3 Report (Previous Reviewer 1)

I have revised the manuscript entitled  “Naturally derived versus synthetic compounds as potential weapons against bacterial biofilms”

Is an interesting and complete revision of Biofilms. Contains very important and fresh information about biofilms and its control.  They added important complementary information. They defended, the kind of manuscript. They are presenting a review an not “critical review” I agree.

Anyway, they mention in the title the world “Compounds” , for this reason, they have to make changes in some parts of the manuscript.  Originally I had made a suggestion in the title “natural strategies” instead “natural compounds”.

One of the most important things in this manuscript is the consideration on the natural control strategies of Biofilms. Liposomes, nanoparticles, the use of bee products, EOs, plant extracts, are very different from "natural compounds". The compounds are simple molecules such as thymol, carvacrol, anthocyanins, ac. coumaric, ac. caffeic etc. Along the manuscript, they mention very few compounds and they focus on the control or natural derived strategies as plants extracts, EOs etc.

I insist in the suggestion of change the title “Natural strategies as potential weapons against bacterial biofilms”. on the other hand, they mention very little comparison with “synthetic” compounds, so I suggest to remove that word from the title.

Whether or not the title change,  This information should be added in the manuscript

Line 428: What kind of virulence attenuation of Pseudomonas? Motility? Adhesion? Mention it.

Table 6: in table 6 there is a column that says “Plant extract”, but  there are also compounds mentioned,. Please add the world compound :  Plant extract, compound or EOs.

Only, If the title do not change, and remains as “compounds”  there are several things that should be improved :

Line 427: Which compounds of Garlic Extract are Quorum sensing inhibitors or Biofilm remover? Mention it.

Line 426-512: You only mention plants extracts, and EOs but very little of the compound implicated, you should mention at least one example of each one.

Line 517-518: You mention Bioactive compounds of Honey. Which ones? Mention it. 

Line 520-525. Honey was used in combination with antibiotics. Was there better inhibition than with the antibiotic alone? was it the same? was minor? Clarify in the manuscript.

Line 548-581. No specific compound is mentioned. Silver nanoparticles obtained by means of different plant extracts are mentioned. It is not mentioned whether any compound was responsible for this antimicrobial or antibiofilm activity. Can the silver nanoparticle be considered as a compound? Are there silver particles loaded with natural compounds? If so, and if the title says "compounds" they should have been mentioned.

Line 708-710 Please remove Remove “However, the emergence of antimicrobial and multidrug resistance  forced the researchers to study all growth features of microbes and their resistance mechanisms for developing effective combating strategies and significant biofilm controlling compounds” Is redundant.

712-717: “In this review, recent studies were reviewed, focusing on biofilm-controlling compounds such as natural plants, bee products”  I insist that they are not compounds, they are “natural strategies”.

Author Response

Reviewer #3

I have revised the manuscript entitled “Naturally derived versus synthetic compounds as potential weapons against bacterial biofilms”

Is an interesting and complete revision of Biofilms. Contains very important and fresh information about biofilms and its control. They added important complementary information. They defended, the kind of manuscript. They are presenting a review and not “critical review” I agree.

Anyway, they mention in the title the world “Compounds”, for this reason, they have to make changes in some parts of the manuscript. Originally, I had made a suggestion in the title “natural strategies” instead “natural compounds”.

One of the most important things in this manuscript is the consideration on the natural control strategies of Biofilms. Liposomes, nanoparticles, the use of bee products, EOs, plant extracts, are very different from "natural compounds". The compounds are simple molecules such as thymol, carvacrol, anthocyanins, ac. coumaric, ac. caffeic etc. Along the manuscript, they mention very few compounds and they focus on the control or natural derived strategies as plants extracts, EOs etc.

I insist in the suggestion of change the title “Natural strategies as potential weapons against bacterial biofilms”. on the other hand, they mention very little comparison with “synthetic” compounds, so I suggest to remove that word from the title.

Dear respected reviewer, the authors agree the idea to change the manuscript title according to your suggestion. The change was operated, please see the new revised version!

Whether or not the title change, this information should be added in the manuscript

Line 428: What kind of virulence attenuation of Pseudomonas? Motility? Adhesion? Mention it.

According to the reviewer requirement, the sentence was rephrased and completed with the requested information, resulting in: „The garlic extract can significantly block QS and may promote rapid virulence attenuation (e.g. elastase, protease A, exo- and cytotoxin production or motility and adhesion capacity reduction) of P. aeruginosa by polymorphonuclear leukocytes (PMNs) within the immune response in a mouse infection model [146].” (lines 427-431 of the revised version).

Table 6: in table 6 there is a column that says “Plant extract”, but there are also compounds mentioned. Please add the world compound: Plant extract, compound or EOs.

The suggested changes by the reviewer was operated resulting in „Plant extracts, Compounds or Essential oils (EOs)” – please see the line 517 of the revised version.

Only, If the title do not change, and remains as “compounds” there are several things that should be improved:

DEAR REVIEWER, THE AUTHORS CHANGED THE MANUSCRIPT TITLE, AS YOU SUGGESTED, SO, THE CONCERNS MENTIONED BELOW, WERE NOT TAKEN INTO CONSIDERATION (IF THE AUTHORS CORRECTLY UNDERSTOOD THE REVIEWER MESSAGE)!

Thank you again for your time and suggestions which have greatly contributed to the manuscript quality!

Line 427: Which compounds of Garlic Extract are Quorum sensing inhibitors or Biofilm remover? Mention it.

Line 426-512: You only mention plants extracts, and EOs but very little of the compound implicated, you should mention at least one example of each one.

Line 517-518: You mention Bioactive compounds of Honey. Which ones? Mention it.

Line 520-525. Honey was used in combination with antibiotics. Was there better inhibition than with the antibiotic alone? was it the same? was minor? Clarify in the manuscript.

Line 548-581. No specific compound is mentioned. Silver nanoparticles obtained by means of different plant extracts are mentioned. It is not mentioned whether any compound was responsible for this antimicrobial or antibiofilm activity. Can the silver nanoparticle be considered as a compound? Are there silver particles loaded with natural compounds? If so, and if the title says "compounds" they should have been mentioned.

Line 708-710 Please remove “However, the emergence of antimicrobial and multidrug resistance forced the researchers to study all growth features of microbes and their resistance mechanisms for developing effective combating strategies and significant biofilm controlling compounds” Is redundant.

712-717: “In this review, recent studies were reviewed, focusing on biofilm-controlling compounds such as natural plants, bee products” I insist that they are not compounds, they are “natural strategies”.

Round 2

Reviewer 3 Report (Previous Reviewer 1)

the authors attended to the corrections. So it can be accepted.

This manuscript is a resubmission of an earlier submission. The following is a list of the peer review reports and author responses from that submission.

Round 1

Reviewer 1 Report

I think this review is very good, well written and although there is another review reported in 2020 by Singh et al. the approach of both manuscripts is very different and different topics are reviewed on in each one. Although both mentioned the QS  there are different points of view. In addition, very important food topics for the industry and the interest of readers are included. 

The only thing I suggest is to change the title to "Natural strategies as a potential weapons against bacterial biofilms" since derived compounds are poorly mentioned, and more extracts and naturraly derived strategies as encapsulation are described. 

Reviewer 2 Report

Comments to the Authors

The manuscript entitled “Naturally derived compounds as potential weapons against bacterial biofilms” has been reviewed. Although the topic may be interesting, the manuscript is not acceptable in the present form.

There is no evidence for a critical assessment of the data, and it remains unclear to the reader what the really salient findings are. Some points are interesting, but Authors have to remember that a review MUST be a critical analysis of the data and do not simply provide a long list of (certainly interesting) facts. In my opinion of reviewer of experience, the review may serve as background material for suggestions of less explored topics in a field, or, on the contrary, as the overview after the conduction of a research project in a specific topic in which the authors have gained state of the art results. The MS is a large list of information, but without a discussion able to point up significative findings.

The material and methods section is missing together with the description of the data mining process, the eligibility criteria and rejection criteria of the cited works, number fo chosen and rejected works. keywords used in the data mining process and in the different databases.

The articles have been selected with a solid and clear methodology? If yes, what? Guidelines for authors clearly recommend PRISMA guidelines.

A paragraph with a complete and clear description of the bibliographic research is mandatory for a review article.

Reviewer 3 Report

The manuscript entitled " Naturally derived compounds as potential weapons against bacterial biofilms" report review on an interesting topic. I suggest that the manuscript could be suitable for publication after minor revision. Thus, I think that the following comments can improve the manuscript.

Comments to authors:

-          The sections “2. Biofilm formation” , “3. Biofilm composition” is better to be shorten and concise.

-          The sections under “8. Biofilm controlling compounds”, more emphasis must be placed on biofilm regulating chemicals, and they must be enriched with more current natural active instances.

-          A better-quality version of Figure 1-6 is required.

-          The references should be updated as many article are published on the topic in 2022 for example:

“Anti-Biofilm and Antibacterial Activities of Cycas media R. Br Secondary Metabolites: In Silico, In Vitro, and In Vivo Approaches’” Antibiotics 2022, 11 (8), 993

-          The references should be revised for missing information.

Reviewer 4 Report

The manuscript reviews phases of biofilm formation and potential weapons against them. An extensive review of the literature has been done, but certain questions and doubts are raised.

1. The title is misleading and does not give the right information about what the content of the text is going to be.

2. The definition of biofilm used in this article needs to be reviewed. e.g. L35-36 '... that eather attached to surfaces or organized into an extracellular matrix.'- is the biofilm on the surface not in the extracellular matrix? Why use the word 'or'? Also in first paragraph of introduction.

3. Are nanomaterials really natural compounds if they are synthesized in labs?

4. From abstract I understood that 'biofilm associated infections' will be the focus - but then the manuscript intertwines medical and food information, it is not clear which MO is the focus (very much is written about P. aeruginosa but then many others appear). It would be easier to follow the text if it was clear what the focus of the article really is. Also, many data are given from simple in vitro studies with no real application to prevent either these infections in humans or say on surfaces.

5. I think the idea of defining the individual phases of the biofilm or stating the structural properties is excellent in connection with the possibilities of influencing them. But at the beginning, you add very informatively which part and how you could influence it. After that, it no longer appears in the text. Supplementing the text with this information would, in my opinion, greatly contribute to the meaning of the manuscript.

6. An extensive review of the literature was done but in the first part very little citation is used and entire paragraphs are written without references. This needs to be corrected.

7. For all control strategies only few examples are listed and no connection to 6.1 or 6.2 are given.

Other comments:

L57: Which infections?

L58, L60: Which antibiotics?

L68: What is 'lethal recurrent'? If it causes lethality it can not happen again?

L68-69: I do not understand this sentence.

L82: Which other molecules?

L93: There is no precise elaboration in Figure 1. These stages are discussed further in text.

L95: Based on what reference?

L97: Text under title '2.1 Adhesion' should be reorganized as it is a bit confusing in the existing format.

L99: Which signaling pathways?

L117-119: Please revise sentence - it is written twice 'adhesion to biotic surface' but only reversible state is listed.

L132: Please change expression 'motile' as dispersion does not happen just because of bacterial motility.

L140: What is dispersal pathway activation?

L163: Pel and Psl are not polysaccharides. It should not be written in italics.

L166: Please revise inclusion of B. licheniformis in this table as 'anti-biofilm role' is probably against other microorganisms and not B. licheniformis.

L166: Please correct how Salmonella is written. Typhimurium is serovar and it is not written in italics and it starts with capital.

L193: Please revise this sentence as it probably does not protect antimicrobials?

L200: What other EPS?

L215: Under subtitles I do not see reference to structural stability.

L218-220: In what way?

L259: To what 'respectively' refers as there are three signaling molecules and two groups of bacteria?

L299: In what way?

L348: Please change word 'efficacious'.

L361 and L371: These paragraph are confusing as are given after lots of general information about biofilm. Please see general comment 4.

L410: Is effect on planktonic cells really the best way to prevent biofilms? It is known from literature that antimicrobials that inhibit bacterial growth can cause negative pressure and possibly add to the development of bacterial resistance.

L420: I am not really sure that this figure contributes to this manuscript.

L473: In this table under 'nanomaterials' are not listed nanomaterials.

L591-592: In the text there is no synthesis of the reviewed literature that will confirm this. Either complete this or delete this sentence.